# An Evaluation of Cultural and Chemical Control Practices to Reduce Slug Damage in No-till Corn

**DOI:** 10.3390/insects13030277

**Published:** 2022-03-11

**Authors:** Galen P. Dively, Terrence Patton

**Affiliations:** Department of Entomology, University of Maryland, College Park, MD 20742, USA; tpatton@umd.edu

**Keywords:** slugs, conservation tillage, residue management, cultural control, baits

## Abstract

**Simple Summary:**

Slugs are the most damaging non-arthropod pest of corn grown in no-tillage systems in the US. The decaying plant residue on the soil surface provides food, shelter and optimum microenvironmental conditions for slug development and survival. In this study, we evaluated several cultural practices to reduce the risk of slug injury and the efficacy of different rates and application patterns of rescue treatments. Corn planted with row cleaner devices to remove surface residue over the seed row and starter fertilizer to enhance seedling growth, together reduced slug activity around emerging plants and provided more favorable conditions for plants to outgrow and tolerate feeding damage. We found that reduced rates of molluscicide baits applied as a directed band over the seed row and broadcasted solutions of urea-based nitrogen applied at night provided effective control as rescue treatments. Practical considerations of these treatments are discussed, as well as changes in weather patterns and current planting practices that have had contrasting effects on slug populations and their potential damage.

**Abstract:**

Slugs, primarily the gray garden slug, *Deroceras reticulatum* (Müller), are the most damaging non-arthropod pest of corn grown in conservation tillage systems in the US. These mollusks favor decaying plant residue on the soil surface, which provides food, shelter and optimum microenvironmental conditions for their development and survival. Here, field plot experiments evaluated several cultural and chemical control practices to suppress slug activity and feeding injury during early seedling growth. The use of row cleaners to remove surface residue over the seed row and starter fertilizer applied different ways during planting significantly reduced the percentage and severity of plants damaged by slugs by negatively affecting their activity around emerging seedlings and providing more favorable conditions for plants to outgrow and tolerate feeding injury. As rescue treatments, reduced rates of a 4% molluscicide bait applied as a directed band over the seed row, and broadcasted solutions of urea-based nitrogen applied under calm winds at night provided effective slug control. Practical considerations of these treatments are discussed, as well as changes in weather patterns and current planting practices that have had contrasting effects on slug populations and their potential damage.

## 1. Introduction

Slugs are the most damaging non-arthropod pest of corn grown in conservation tillage systems in the US [1,2,3,4,5,6]. Although several species have been associated with corn (*Zea mays* L.), the gray garden slug, *Deroceras reticulatum* (Müller), is the most abundant mollusk causing economic damage to germinating and seedling stages during prolonged moist, cool periods of the spring, especially in continuous no-tillage fields [4,6]. Slugs favor areas that receive sufficient moisture to keep them comfortable, and that have an abundance of decaying plant residue on soil surfaces, which provides food, shelter and optimum microenvironmental conditions for their activity, development, and survival. High perennial slug populations are particularly common in no-tillage production areas around the Chesapeake Bay, where conservation practices require growers to maintain a certain amount of plant residue on the soil surface year-round [7].

Slugs damage corn shortly after germination by feeding belowground on the seed and coleoptile or later by scraping strips and shredding leaves of seedlings, killing plants or retarding growth [2,6,8]. The most severe feeding injury occurs when moist soil conditions prevent the seed furrow slot from closing completely during planting, enabling slugs to feed directly on the apical meristem below the soil surface. This results in plant stand losses which often requires re-planting. Older corn seedlings are able to tolerate a considerable amount of slug injury before corn growth and yield are severely affected, provided that weather conditions are favorable for growth [9,10]. In a heavily infested no-till corn field, preliminary observations assessed the ability of corn seedlings in the two- to three-leaf stage to recover from slug injury. Two weeks after initial feeding, 38% of severed seedlings and 83% of seedlings with 50–75% defoliation regrew and recovered from the injury, while all seedlings with <50% injury recovered completely to normal growth (unpublished data, GPD). Given this ability to recover from heavy slug damage, any cultural practices that enhance seed germination and early seedling growth may function as effective management tactics by shortening the window of opportunity for slugs, thus mitigating losses resulting from early feeding injury.

The most effective cultural practice to minimize losses caused by slugs is simply to change tillage practices. To reduce the surface residue, many growers with perennial slug problems have had to apply minimum tillage using a shallow chisel plow or disk for at least one growing season; however, this tactic is not always possible if growers prefer to avoid tillage or are locked into continuous no-tillage cropping systems in accordance with conservation practice standards [7]. An alternative tactic is the use of row cleaner devices on corn planters to push corn stalks and other plant residue aside, thus preparing a 20 to 25 cm zone of relatively clean soil surface in front of the double-disc opener. The cleaned surface area allows the seedbed to warm up faster, encouraging rapid seed germination and seedling growth, and reduces seed germination problems resulting from pieces of residue being jammed into the seed slot over the seed [11,12,13]. Another cultural practice that can enhance corn seedling vigor and rapid growth is the use of starter fertilizer at planting [14,15]. This practice is particularly recommended for early planted no-till fields with high residue cover that experience cooler soil temperatures [16]. Most growers in the mid-Atlantic US apply at planting a portion of the total nitrogen and phosphorus requirement for corn. Common methods of application include in-furrow application, placement alongside the seed slot, and banded application over the seed slot [17]. Together, the extra boost in growth resulting from row cleaners and starter fertilizer has been suggested to help corn seedlings tolerate and outgrow slug injury [6]; however, their effects on slugs have not been empirically tested.

Molluscicides are widely used in Europe and other countries where there are more serious and consistent slug problems [18]. In the US, despite the increasing importance of slugs in no-tillage cropping systems, there has been less commercialization of molluscicides, mainly because of the sporadic and localized nature of slug injury in crop fields and the potential environmental risks associated with some of the toxic active ingredients. Metaldehyde formulated as a pelleted bait is the most widely used molluscicide in corn (and other crops) but it is expensive, difficult to apply and dispense evenly, and has inconsistent efficacy when cool, rainy conditions follow application, reducing the residual activity and toxicity of the bait [18,19,20,21]. Other bait products containing iron phosphate are also labeled for use in corn but provide more variable control. Current labels of metaldehyde baits registered for use in corn recommend only broadcast applications at planting before slug damage occurs. However, it may be effective and more economically feasible to apply baits as a banded application directed over the seed row where slugs are active. In the same way, preliminary observations suggest that the salt content of certain dry and liquid forms of nitrogen and potassium applied over the row may have negative effects on slug behavior and survival. For instance, cursory evidence based on grower testimony and extension information [22] suggests that a broadcast treatment of urea-based nitrogen solution applied at night when slugs are actively feeding on plants can be an effective rescue option for controlling localized infestations in corn fields. The rationale is that the nitrogen acts as a contact poison and irritant to slugs, while the cost of the application is partly offset by the added fertility. However, the proper timing, placement, and control efficacy of these fertilizer-based control options have not been experientially evaluated.

In this study, we investigated cultural practices used at corn planting that could reduce the risk of slug damage and tested several rescue control options. The specific objectives were: (i) to assess the effects of residue management using row cleaners to suppress slug activity and feeding injury; (ii) to determine if fertilizers can be applied in a manner that will either increase the ability of corn seedlings to tolerate slug injury or negatively act directly on slugs; (iii) to quantify the effectiveness of different rates and application patterns of molluscicide baits; and (iv) to evaluate the control efficacy of a broadcast application of urea-based nitrogen as a rescue treatment.

## 2. Materials and Methods

### 2.1. General Description of Field Sites

Experiments were conducted during 1992–1994 on two University of Maryland Research and Education Centers located at Wye on the Eastern Shore of Maryland (WREC) and at Clarksburg in central Maryland (CREC). All fields were previously planted in a soybean-corn rotation or continuous corn system using no-tillage practices for at least 5 years. Each field had a history of slug activity associated with a heavy layer of surface residue. Plots at all sites were no-till planted during early to mid-May with Pioneer 3394 corn seed in 76 cm spacings at a target population of 73,000 plants/ha. We applied a broadcast application of 0-15-30 fertilizer at least one week prior to planting and side-dressed with urea-based N around four weeks after planting according to the specific nutrient requirements at each site. The side-dress rate was adjusted for the amount applied as experimental treatments so that the total N applied was the same across all plots. Standard burndown and residual herbicides were applied prior to planting and post-emergent herbicide were used if necessary. An insecticide was also applied with the burndown herbicides to ensure insect-free damage during the early stages of corn growth. The corn seed was fungicide-treated and no foliar insecticides were applied after planting. Plots at several sites were machine harvested with a Massey Ferguson 8XP plot combine.

### 2.2. Experiments Testing the Effects of Row Cleaner and Row Fertilizers

Treatment plots in each experiment were arranged in a randomized block factorial design consisting of two whole plots (with and without row cleaner), each split into two or more fertilizer treatments, depending on the site/year. Surface residue was managed with a finger-wheel row cleaner/coulter blade combination [13] mounted in front of the seed opening discs and adjusted to clear surface residue over the seed row. The addition of a coulter blade not only helped to cut the residue but also penetrated and loosened the soil to aid in proper seed placement and good seed-to-soil contact [13].

At WREC (1992), eight replicate blocks of whole plots were planted on May 1, each measuring 12 rows running 45 m, and divided into two subplots of six rows each. One subplot received no starter fertilizer as the control, while the other received 30 kg N/ha of 34-0-0 fertilizer applied at planting as a deep-band 5 cm to the side and 5 cm below the seed (herein referred to as 5/5SF). This represented the most common placement of starter fertilizer used in no-till corn at the time in the mid-Atlantic US. At CREC (1993), four replicate blocks of whole plots of 12 rows were planted on 12 May and split into three subplots, each measuring 4 rows 15 m long. Subplot treatments included: (1) 30 kg N/ha as a 5/5SF application; (2) broadcast application of 10-10-10 at 500 kg/ha applied prior to planting; and (3) no starter fertilizer as the control. At WREC (1993), the experiment was planted on May 8 and arranged as an incomplete split-split plot design comprising three replicate blocks, each with whole plots planted with and without row cleaners. Each whole plot was split into either two or four subplots, each measuring 8 rows 15 m long. Subplot treatments included: (1) no starter fertilizer as a control; (2) 30 kg N/ha as a 5/5SF application; (3) in-furrow application of ammonium nitrate at 10 kg N/ha; and (4) banded application of 25 kg/ha of ammonium nitrate over the closed seed slot. Subplot treatments 1 and 2 were arranged in each whole plot planted with row cleaners, while all four treatments were arranged in each whole plot without row cleaners. Each row cleaner-fertilizer combination was further divided into two sub-subplots receiving applications of potash at 120 kg K_2_O/ha either banded over the seed row or broadcasted at the spike stage by manually spreading weighed amounts over each row. These treatments were included to determine if the salt content of the potash fertilizer negatively affects slug activity.

Sampling methods and measured variables to assess treatment effects varied slightly among experiments. In all experiments, the number of slugs on plants, percentage of damaged plants, and severity of feeding damage were assessed at two to three weeks after planting by visually examining 20 consecutive plants from a center row of each plot. Damage severity was rated as follows: 1 = undamaged plant; 2 = only one or two older leaves with light to moderate injury; 3 = leaves less than 50% damaged, intact and displayed normally; 4 = leaves more than 50% shredded apart but still connected, most recent one displayed normally; and 5 = plant mostly severed at base, with few leaves displayed. To assess treatment effects on seedling growth and survival, we recorded the plant population density per 15-meter row, and the number and extended length of leaves averaged over 20 plants at the 4–6-leaf stage. In both 1993 experiments, corn yields were machine-harvested and adjusted to a standard moisture level.

### 2.3. Experiments Testing Different Rates and Application Patterns of Molluscicide Baits and Urea-Based Nitrogen Sprays

Separate experiments were conducted at CREC and WREC to evaluate the effectiveness of metaldehyde bait for the control of slugs. Plots were established in heavy slug-infested sections of the same 1993 sites described above but not overlying with other studies. Plots measuring 8 rows 15 m long were arranged in a randomized-complete block design with four replications. Each experiment included an untreated control plot planted side by side with plots of different rates and application patterns of a 4% metaldehyde bait (Deadline Bullet, AMVAC Chemical Corp., Los Angeles, CA, USA). This formulation was the same as the Deadline Bullet and M-Ps products currently labeled at the rate of 25 kg/ha for use in corn, except that several pellet design changes have since been made to improve spreading and durability characteristics. Experiments in 1993 tested the 10 kg/ha rate banded over the seed row and a higher rate of 40 kg/ha as a broadcast application at the spike stage about 2 weeks after planting. In 1994, the same CREC and WREC fields were planted again in no-till corn on 13 May and 10 May, respectively, and paired treated and control plots were arranged to evaluate lower rates of 5 and 10 kg/ha of bait applied at the spike stage as a narrow band over the seed row. Baits were applied only once after the majority of eggs had hatched but before extensive slug injury occurred on the seedlings. Each rate was delivered with a small spreader (model GT-77, Herd Seeder Company, Inc, Logansport, IN, USA) mounted to an ATV for broadcast application or by manually spreading weighed amounts evenly in a 25 cm band over individual seed rows. At WREC (1994), two potash treatments were also tested for their direct effects on slug activity by broadcasting 120 kg K_2_O/ha at planting and at spike stage. For all site years, we recorded the number of slugs, percentage of damaged plants, and severity of feeding damage at 4 weeks after planting by visually examining 20 consecutive plants from a center row of each plot.

At WREC in 1994, foliar applications of liquid N solution applied at night on 3 June were evaluated as a rescue treatment in a different section of the same field. Three rates of 41.6, 83.3, and 166.5 L/ha of urea-ammonium nitrate fertilizer were applied at night around 11 PM when winds were calm and slugs were actively feeding on plants. Treatments were applied using a tractor-driven boom sprayer delivering in 208.2 L/ha of diluted spray over smaller plots of 4 rows 15 m long. Because slug densities varied across the field, each treatment rate was replicated 16 times and compared directly with adjacent untreated plots arranged in a randomized split-plot design. Treatments were applied at the 3- to 4-leaf stage and then evaluated the following night under similar weather conditions between the hours of 11 PM and 2 AM to record the number of slugs per 20 plants from a center row of each plot. We also collected additional data the next day on the severity of feeding damage expressed as percent defoliation and the percentage of residue covering the surface of each untreated plot.

### 2.4. Data Analysis

Before analysis, data recorded from subsamples of plants per replicate unit were averaged. We then tested each data set for normality and homogenous variance using the Shapiro–Wilk *W* test, Spearman’s rank correlation, and by examining residual scatter plots. For each variable, we performed data transformations prior to analysis and partitioned the variance, if necessary. The WREC (1992) and CREC (1993) experiments were analyzed separately to test for main and interaction effects of the row cleaner and starter fertilizer treatments, using a Proc Mixed ANOVA [23], with replicate block treated as a random effect. Due to the incomplete design of the WREC (1993) experiment, treatment effects at the whole plot, subplot and sub-subplot levels were analyzed separately as a one-way ANOVA. Variables recorded from the molluscicide bait and urea-based nitrogen experiments were analyzed as a matched-pair randomized ANOVA. When slug counts were recorded, percent control was expressed as a reduction in slug density relative to the density in the paired untreated plot. In all analyses, the Student–Newman–Keuls method was used to test for significant among multiple means at the 5% probability level. All summarized results were displayed as the mean (±SEM) of each variable.

## 3. Results and Discussion

Slug densities and the timing and extent of their damage relative to corn seedling growth varied among experiments. At WREC (1992), percentage of damaged plants at the two-leaf stage averaged 16.6 ± 2.83% in plots without residue management and starter fertilizer. Although not part of the experimental design, it is noteworthy that slug damage averaged 2.0 ± 0.02% in adjacent sections of the same field where minimum-tillage with a chisel plow and disc was used during the previous fall. Slug activity was much higher in untreated control plots in 1993, with the percentage of damaged plants at the two-leaf stage averaging 71.4 ± 9.09% and 79.3 ± 6.13% at CREC and WREC, respectively. Likewise, plants damaged by slugs in the same fields in 1994 averaged 71.4 ± 0.02% at CREC and 48.1 ± 4.53% at WREC. Surface residue cover, weather conditions, and the timing of egg hatch of slugs related to seedling growth in the spring were likely factors contributing to the different levels of slug damage. For instance, the 1992 field was planted one week earlier than the other sites and exposed to cooler temperatures during early seedling growth. Our observations indicate that peak egg hatch occurred after seedlings reached the two-leaf stage, when warmer temperatures were less favorable for slug activity and enabled plants to outgrow the feeding injury. At all sites, counts of slugs active on plants during the day were highly variable among plots and not always indicative of the percentage and severity of damaged plants. The highest slug density was recorded at night in untreated plots of the 1994 field at WREC, which averaged 19.8 ± 4.45 slugs per 20 plants. The amount of slug damage in these plots was directly related to the level of surface residue cover. Based on visual estimates, percentage defoliation averaged 8.1, 18.2, 29.2 and 44.9% in plots with <40%, 40–60%, 61–80%, and >80% residue cover, respectively.

### 3.1. Effects of Row Cleaners and Row Fertilizers

With few exceptions, the interaction effect of row cleaner and row fertilizer was not significant for most variables recorded. However, when the interaction was significant, differences changed in magnitude but remained relatively ranked in the same order between the two factors. Thus, experimental results are mainly explained by the main effects. In all experiments, visual estimation of the row cleaner treatment removed more than 80% of the plant residue from a 20–25 cm wide zone on the surface of the seed row, which had a significant negative effect on slug activity and feeding damage. In the 1992 experiment, row cleaning significantly reduced slug damage by 48.4% (F_1,21_ = 11.86, *p* = 0.002) and resulted in a small but significant increase in the final plant population (F_1,14_ = 5.19, *p* = 0.039) (Figure 1). Subplots treated with 5/5SF fertilizer also had 28.0% less slug damage but the difference from the control was not statistically significant. However, starter fertilizer had a significant positive effect on seedling growth (extended leaf length: F_1,14_ = 118.37, *p* < 0.001). Altogether, the combined effects of planting without row cleaning and starter fertilizer resulted in the lowest plant population (F_1,14_ = 13.53, *p* = 0.003) and the highest level of slug damage.

Removal of surface residue from the seed row had fewer overall effects on the higher slug populations experienced at both field sites in 1993. At CREC, there was 13.9% and 12.2% less damaged plants in plots planted with row cleaners and 5/5SF fertilizer, respectively, but differences were not statistically significant (Figure 2). However, row cleaning created a cleaner, more environmentally favorable seedbed to allow proper seed placement and plants to emerge more uniformly, as evident by the final plant population (F_2,21_ = 9.89, *p* < 0.001) and corn yield (F_2,21_ = 10.14, *p* = 0.005) which were 7.9% and 16.1% higher in the row cleaner plots, respectively. The fertilizer treatments also affected the growth of seedlings and their ability to outgrowth further slug feeding. The 5/5SF significantly reduced the severity of injury (F_2,12_ = 9.55, *p* = 0.003) and both fertilizer treatments enhanced seedling growth by an overall 16.0% (extended leaf length: F_2,12_ = 16.36, *p* < 0.001). At WREC, the row cleaner effect was significant (F_1,6_ = 6.46, *p* = 0.044), resulting in a 29.4% reduction in the percentage of damaged plants (Figure 3). There was also evidence of reduced damage severity (*p* = 0.090) and higher plant population and yields but differences were not statistically significant. The fertilizer effect was highly significant for the percentage of damaged plants (F_16,60_ = 21.77, *p* < 0.001), primarily due to the in-furrow and banded applications of ammonium nitrate, which reduced slug damage by 65.0% and 45.2%, respectively. Additionally, both treatments significantly increased the final plant population by 5.3% to 8.0% (F_3,30_ = 5.82, *p* = 0.003). We expected that the salt content of the potash fertilizer would negatively affect slug activity, but both applications had no significant impact on any of the measured variables.

Despite the relative differences among experiments, overall results indicate that removal of surface residue over the seed zone at planting improves corn emergence and can help to reduce slug damage in no-tillage systems. Slugs are very sensitive to slight changes in soil moisture and need protective cover to survive higher temperatures during the day [8,24,25]. Studies have shown that cleaning residue from the seed zone increases soil temperature by 1.5 to 3 °C, resulting in faster germination and emergence of the corn seedlings [26,27]. Removal of residue also reduces the soil moisture in the cleared zone [28]. Consequently, the microenvironmental conditions surrounding the seed and emerging plants were less favorable for slug activity. Furthermore, faster germination and plant emergence enhanced the ability of corn seedlings to outgrow and tolerate slug injury, especially under cool and moist spring soil conditions typical of no-tillage systems. However, it is noted the plant growth in plots without residue management eventually equaled and sometimes surpassed the growth in plots planted with row cleaners. Nonetheless, the improved seedbed and less slug injury resulted in plant populations 8 to 10% higher in plots planted with row cleaners.

The use of starter fertilizer at planting also had an indirect influence on the amount of slug damage, mainly by the enhancement of germination and seedling growth, which effectively reduced the period of vulnerability to slug feeding. Studies has shown that application of fertilizer close to the seed at planting increases seedling growth and plant height [14,17,29,30,31]. Since seedling growth was similar among the three fertilizer applications in the WREC study (Figure 3), it is interesting that the in-furrow and banded applications of ammonium nitrate at planting had significantly greater impact on slug damage than the 5/5SF application. These treatments had more direct contact with slugs, even the in-furrow application which deposited a portion of the fertilizer on the surface around the seed slot. Thus, it is likely that the salt and acid effects of ammonium nitrate acted as a direct irritant and impeded slug activity. Yet, it is unclear why the potash treatments, applied later in seedling growth, had little effect on both seedling growth and slug damage. These fertilizer responses on slugs warrant further investigation.

### 3.2. Effects of Molluscicide Baits and Urea-Based Nitrogen Sprays

Results of the Deadline Bullet bait treatments show a range of effects on slug damage and other measured variables depending on the rate and application method used. At CREC (1993), the number of slugs on plants during the day in untreated plots averaged 6.4 ± 1.83 per 20 plants. The effectiveness of baits at 10 kg/ha banded over the seed row and 40 kg/ha broadcasted was not statistically different but both rates reduced the overall percentage of damaged plants (F_2,14_ = 18.7, *p* < 0.001) and severity of feeding damage (F_2,14_ = 223.5, *p* < 0.001) by 37.6% and 42.8%, respectively (Figure 4). Plants were already showing some slug damage when the rescue treatments were applied at the spike stage, so the baits effectively prevented further slug feeding. This protection resulted in significantly greater seedling growth (extended leaf length: F_2,14_= 5.77, *p* = 0.015) and a higher stand population (F_2,14_ = 3.79, *p* = 0.048). At WREC (1993), slug numbers observed during the day were lower (mean 1.81 ± 0.64 per 20 plants) and more variable across untreated plots. Plant damage in the baited plots was not significantly different from the untreated control (Figure 4); however, both treatment rates significantly reduced the severity of damage (F_2,4_ = 60.6, *p* = 0.001).

Since the 10 kg/ha rate of Deadline Bullet gave near similar levels of slug control compared to the high rate in 1993, we further tested 5 and 10 kg/ha rates banded over the row in 1994. Slug densities reported during the day in untreated plots averaged 12.0 ± 1.17 and 6.6 ± 0.88 per 20 plants at CREC and WREC, respectively. Rates of 5 and 10 kg/ha significantly reduced the percentage of damaged plants by 31.8% and 43.2% at CREC (F_2,22_ = 27.4, *p* < 0.001) and by 68.1% and 76.7% at WREC (F_4,115_ = 12.2, *p* < 0.001), respectively (Figure 5). Although the higher rate always appeared more effective, differences between rates at both sites were not significant. The 5 and 10 kg/ha rates also significantly reduced the damage severity rating by 38.2% and 47.4% at CREC (F_2,29_ = 139.9, *p* < 0.001) and by 36.2% and 38.9% at WREC (F_4,115_ = 7.6, *p* < 0.001); and the ratings at CREC were significantly different between rates (Figure 5). The additional dry granular treatments of 120 kg/ha of potash at WREC had no overall effects on any of the measured variables, except that the percentage of damaged plants was significantly reduced by the at-planting application. In the same field at WREC where slug densities reached an overall mean density of 19.7 ± 4.46 slugs per 20 plants at night, rescue treatments of urea-ammonium nitrate solution applied at night significantly reduced the slug population (F_2,33.5_ = 7.14, *p* = 0.003) (Figure 6). Rates of 41.6, 83.3 and 166.5 L/ha reduced slug densities by 38.9%, 77.3% and 94.9%, respectively. At the time of treatment, weather conditions of around 9 °C. and high humidity with calm winds were ideal for slug activity. No phytotoxicity was observed, except for some minor leaf burn at the higher rate.

Overall, these experiments demonstrate that there are effective ways to apply rescue treatments to prevent heavy damage by slugs in no-till corn. Our results agree with other studies reporting greater than 80% control of slugs from broadcast applications of 4% metaldehyde pelleted products if timed properly [32,33,34,35]. However, as a more cost-effective option, lower per ha rates applied as a directed band over the seed row can provide enough protection to prevent further slug damage. In effect, this approach increases the baiting points available to slugs where they are actively feeding. For example, a broadcast application of Deadline Bullet or M-Ps at 25 kg/ha delivers approximately 118 pellets/m^2^, whereas a banded application at 10 kg/ha over a 20 cm zone of the seed row delivers 180 pellets/m^2^. For effective control, timing of a molluscicide application should coincide after peak egg hatch when juvenile slugs are present [35]. Although not routinely done, slug control decisions can be determined by monitoring slug populations prior to or after planting by visual counts at night or refuge traps [6,36,37]. Certainly, at-planting bait applications as a preventive option would be more operationally feasible with corn planters equipped with granular applicators and could possibly eliminate a later trip over the field. However, molluscicide baits have relatively short residual activity and thus may not be effective when slugs are later feeding on emerging plants. Our results suggest that a better approach is to apply a banded molluscicide after plants have already emerged when there is evidence of early feeding injury to make prescription-based control decisions. Although this reduces the cost of the bait, practical considerations need to be evaluated, especially the operational feasibility and cost of effectively applying a uniform banded treatment of a pelleted bait after planting. Moreover, regardless of how and when baits are applied, they will likely be ineffective if slugs are feeding below the surface in improperly sealed seed slots.

An even more effective and practical rescue option that can be applied with standard sprayer equipment is the broadcast treatment of urea-ammonium nitrate solution at night when slugs are feeding on plants. The 83.3 L/ha rate in diluted spray is probably the best choice because of the reduced cost and less risk from direct injury to the corn plants. However, this rescue option depends on whether slugs are present on plants when the foliar treatment is applied. Inconsistencies in control efficacy of applications at night have been largely correlated with windy conditions and cold temperatures which can drastically reduce slug movements and activity on plants. Although the cost of an over-the-top application of urea-based nitrogen may be partly offset by the added fertility, further studies are needed to determine how much N actually becomes available for plant growth. Accordingly, when using this rescue option, growers may have to adjust the side-dressing rates of N to keep in compliance with nutrient management requirements in many states.

## 4. Conclusions

Slugs are a major pest problem in conservation cropping systems in the US, but certain cultural practices used at planting can be manipulated to make conditions unfavorable for slug activity, development, and survival. This study provides clear evidence that planting corn (and likely soybean as well) with row cleaners and starter fertilizer aids in mitigating the risk of slug damage by reducing slug activity around emerging plants and by providing more favorable conditions for plants to outgrow and tolerate feeding damage. However, these cultural practices alone may not prevent economic losses in fields with high levels of slug activity. As rescue treatments, our results show that reduced rates of 4% metaldehyde pelleted baits applied as a directed band over the seed row can provide effective control at a lower cost if properly timed to slug activity but, as discussed, there exist concerns over the operational practicality of their use. A more practical rescue treatment is a diluted solution of urea-based nitrogen broadcasted at night as a contact poison and irritant to slugs feeding on plants; however, calm winds and optimum temperatures at night are required for successful slug control. Using a similar approach, potash fertilizer applications tested for their direct effects on slugs resulted in mixed results.

In summary, we acknowledge that this study was conducted years ago; however, the objectives have not been empirically addressed and the results are still relevant to the ongoing challenges of managing slugs as more growers adopt no-tillage practices. Yet, since this study was conducted, there have been changes in weather patterns and planting practices that will likely have contrasting effects on slug populations and their potential damage to crops. First, milder winters due to climate change should favor higher survival of overwintering slug stages and an earlier hatch of juveniles coinciding with more vulnerable seedling growth stages. Moreover, climate change is expected to increase precipitation in the mid-Atlantic US which will also favor slug activity and survival. Second, the majority of no-till corn and soybean in the mid-Atlantic region is planted into cover crops killed with herbicide prior to planting; and some growers are beginning to “plant green” and terminate the cover crop before the crop emerges. There is growing evidence that these practices may help to reduce slug damage by supporting higher populations of slug predators [38,39] and also diverting slugs away from feeding on emerging crop seedlings [40,41]. However, more research is needed to evaluate their benefits for managing slugs in combination with the cultural practices evaluated in this study. Conflicting with these benefits, several studies have shown that the prophylactic use of insecticide-treated corn seed and certain insecticides applied at planting have indirectly enhanced slug populations by negatively affecting their predators via prey-mediated exposure [42,43,44]. Lastly, the use of row cleaners has declined over the years due to equipment maintenance costs and the fact that removal of surface residue in fields with heavy cover crops has become more problematic and can also result in weed problems over the seed row. As our results show, planting without row cleaners in fields with heavy residue cover can potentially increase the risk of slug problems. Nevertheless, developments in no-till planter technology have reduced the worst-case losses caused by slugs feeding below the surface in improperly sealed seed slots. Growers are now using heavier corn planters with improved closing wheels and downforce mechanisms to ensure the seed furrow is closed under the specific soil condition encountered.

## Figures and Tables

**Figure 1 insects-13-00277-f001:**
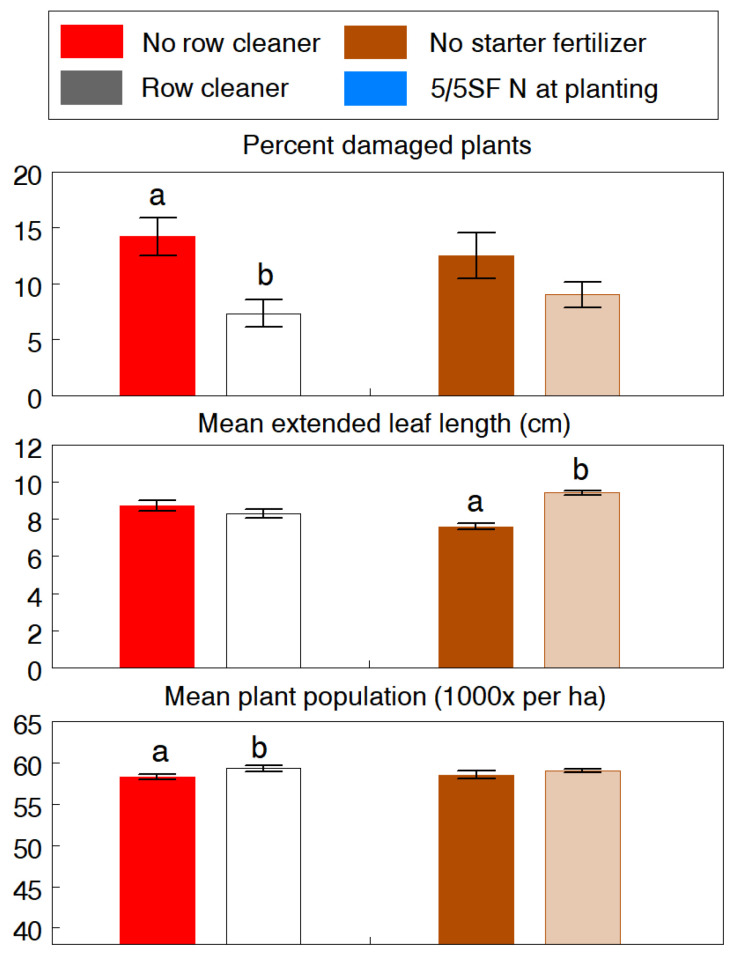
WREC experiment (1992). Effects of using row cleaner and starter fertilizer practices at planting on damage caused of the gray garden slug, *Deroceras reticulatum*, and on the seedling growth and population in no-tillage corn. The 5/5SF treatment was 30 kg N/ha of 34-0-0 applied as a deep-band 5 cm to the side and 5 cm below the seed. Pairs of means (±SEM) of each main effect with different letters are significantly different at the 5% probability level.

**Figure 2 insects-13-00277-f002:**
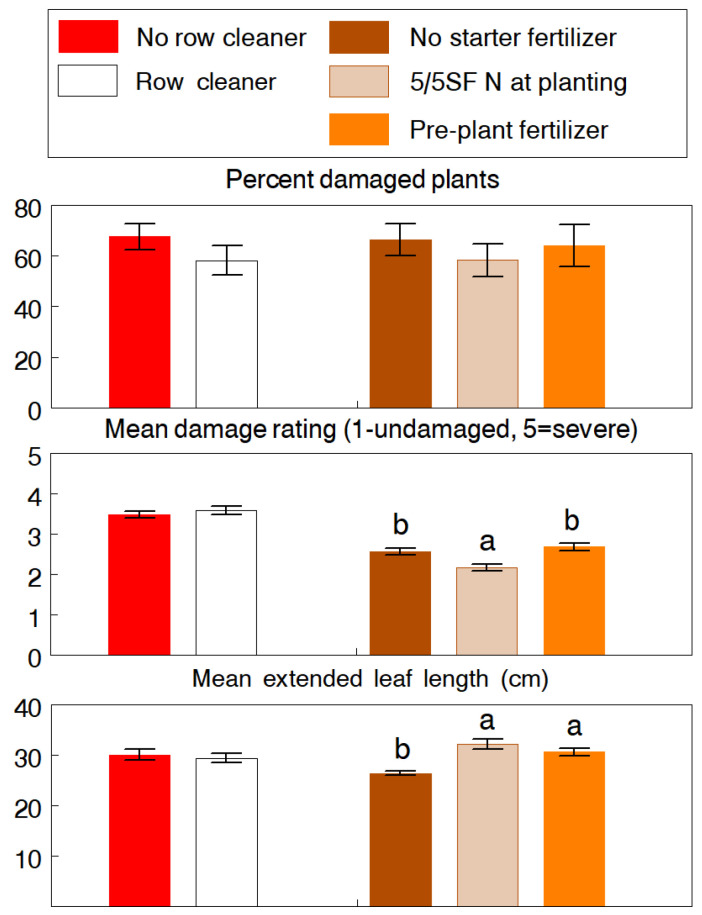
CREC experiment (1993). Effects of using row cleaner and different starter fertilizer practices at planting on damage caused of the gray garden slug, *Deroceras reticulatum*, and on the seedling growth and plant population in no-tillage corn. The 5/5SF treatment was 30 kg N/ha of 34-0-0 applied as a deep-band 5 cm to the side and 5 cm below the seed. The pre-plant treatment was a broadcast application of 10-10-10 at 500 kg/ha applied prior to planting. Pairs of means (±SEM) of each main effect with different letters are significantly different at the 5% probability level.

**Figure 3 insects-13-00277-f003:**
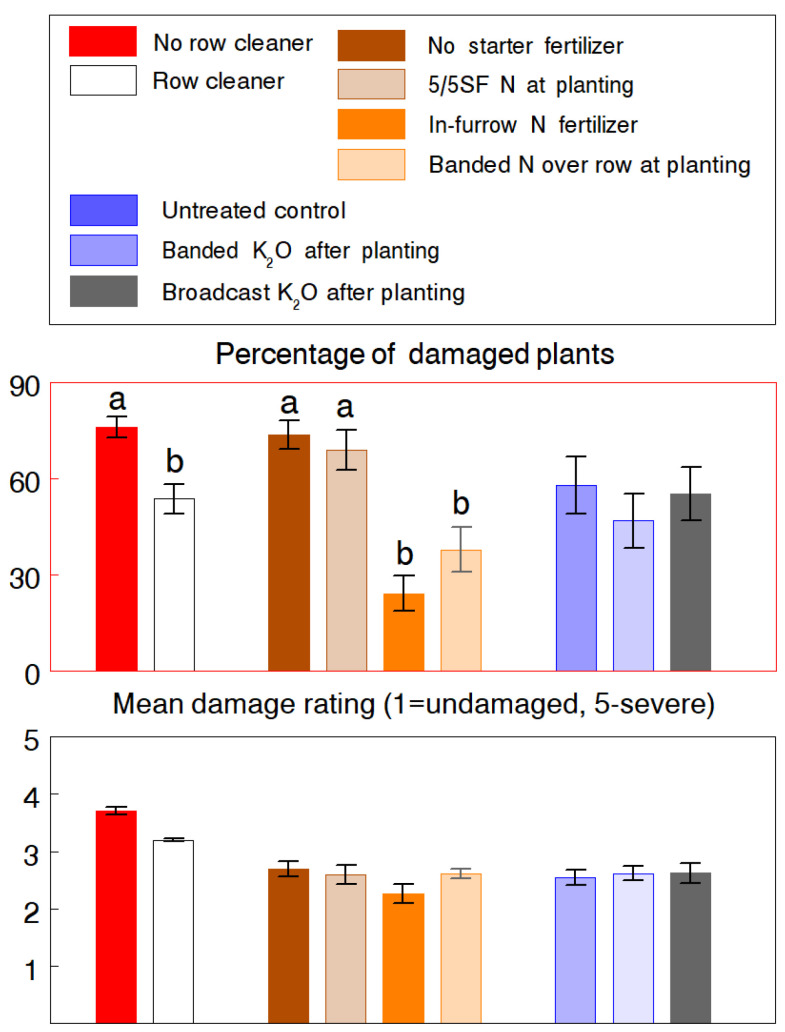
WREC experiment (1993). Effects of using row cleaner, different starter fertilizer practices, and dry granular potash treatments on damage caused of the gray garden slug, *Deroceras reticulatum*. Fertilizer treatments were: (1) 5/5SF of 30 kg N/ha of 34-0-0 applied at planting as a deep-band 5 cm to the side and 5 cm below the seed; (2) in-furrow application of ammonium nitrate at 10 kg N/ha; and (3) banded application of 25 kg/ha of ammonium nitrate over the closed seed slot. Application of potash at 120 kg K_2_O/ha was either banded over the seed row or broadcasted at the spike stage. Pairs of means (±SEM) of each of the three main effects with different letters are significantly different at the 5% probability level.

**Figure 4 insects-13-00277-f004:**
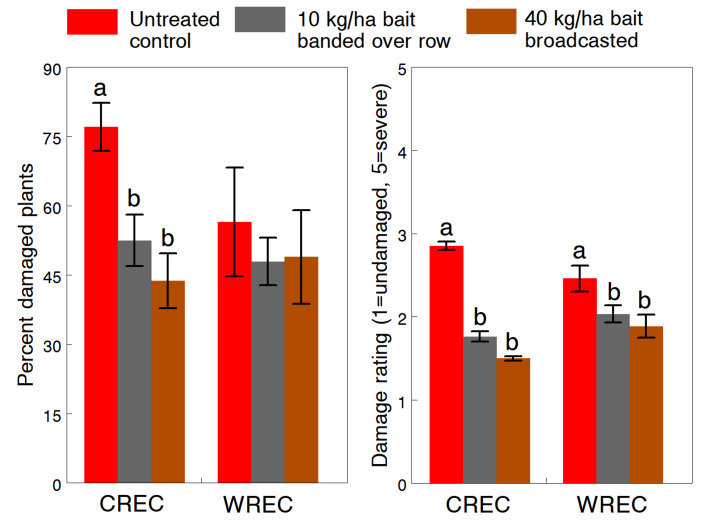
Effects of banded and broadcast applications of pelleted 4% metaldehyde bait (Deadline Bullet, AMVAC Chemical Corp., Los Angeles, CA, USA) on the percentage and severity of plant damage of the gray garden slug, *Deroceras reticulatum*. Means (±SEM) are presented for studies conducted at two University of Maryland research farms in 1993. Pairs of means within each study with different letters are significantly different at the 5% probability level.

**Figure 5 insects-13-00277-f005:**
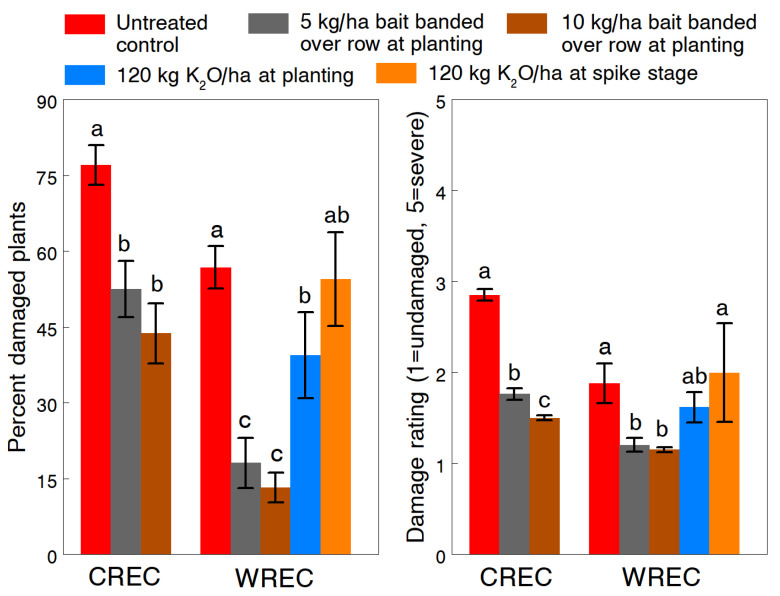
Effects of banded applications of pelleted 4% metaldehyde bait (Deadline Bullet, AMVAC Chemical Corp., Los Angeles, CA, USA) and broadcast dry granular applications of potash on the percentage and severity of plant damage of the gray garden slug, *Deroceras reticulatum*. Means (±SEM) are presented for studies conducted at two University of Maryland research farms in 1994. Pairs of means within each study with different letters are significantly different at the 5% probability level.

**Figure 6 insects-13-00277-f006:**
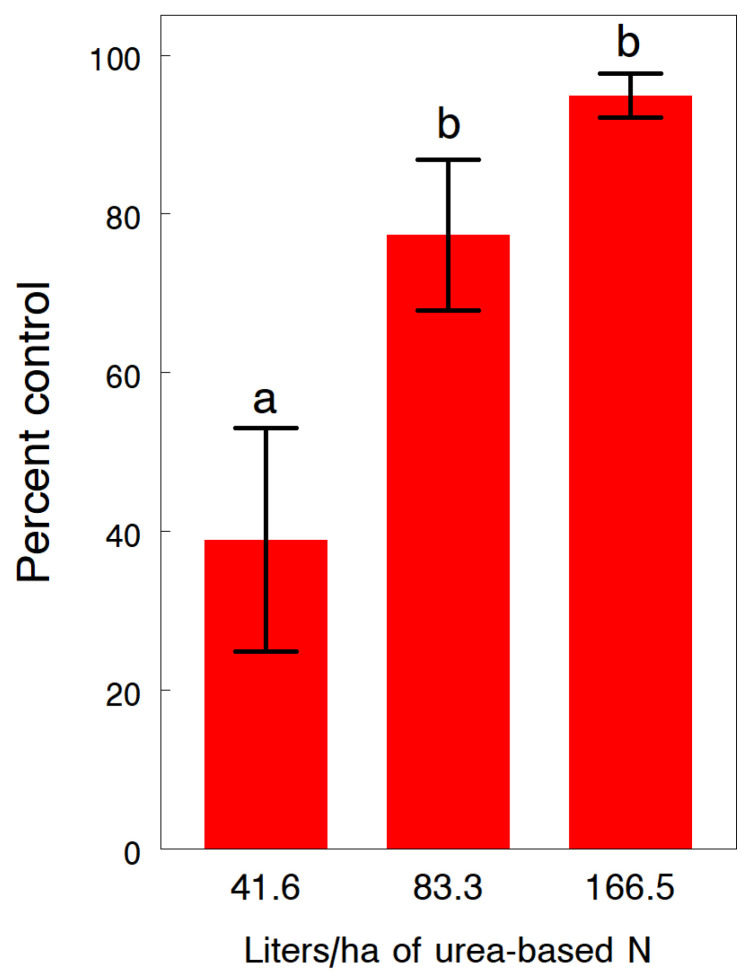
Effects of three rates of urea-based ammonium nitrate solutions applied at night to reduce the percentage and severity of plant damage of the gray garden slug, *Deroceras reticulatum*, in no-till corn. Treatments were applied at the three- to four-leaf stage and then evaluated the following night between the hours of 11 PM and 2 AM to record the number of slugs per 20 plants from a center row of each paired treated and untreated plot. Percent control was expressed as a reduction in slug density relative to the density in the paired untreated plot. Pairs of means (±SEM) with different letters are significantly different at the 5% probability level.

## Data Availability

The data presented in this study are available on request from the corresponding author. The data are not publicly available due to privacy restrictions.

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
