# Peer review of "An Evaluation of Cultural and Chemical Control Practices to Reduce Slug Damage in No-till Corn"

_insects, 2022, doi:10.3390/insects13030277_

Round 1

Reviewer 1 Report

General slug related questions

How wide should a slug plot be?

How long would you say between soil disturbance and an increase in slug population?

Specific questions to the manuscript:

May want to briefly mention iron phosphate baits in the introduction such as Sluggo, Iron Fist, and Ferroxx. I concede that efficacy is pretty variable compared to Deadline.

Please include planting date for all experiments.

With the Urea applications, do you have temperature, relative humidity and date of application?

Do you have an estimate for proportion of D. laeve? You mention timing of egg hatch of slugs related to seedling growth was likely a factor contributing to different levels of slug damage, with the 1992 field planted earlier and with ‘observations indicate peak egg hatch occurred after 2-leaf stage’. This is similar to an anecdote in Ron Hammond’s 1996 slug bait paper. Can you elaborate any more on this? Do you have data suggesting what date peak egg hatch occurred near? I start finding gray garden neonates around the first week of April, but D. laeve anytime in the spring and late winter.

Can you confirm that the highest slug density was 1994 WREC even with 2nd lowest slug injury to plants?

The methods indicate that plots were harvested with a Massey Ferguson, but there is no indication of yield data in the results. If you do not intend to record yield, perhaps it would be better to remove that line.

Any thoughts as to why in-furrow reduced slug damage? Better/faster plant growth? Were seed slots closed?

In the discussion – rephrase the sentence “Our results show that better approach is to apply….after plants have already emerged.” A pre-plant vs post-emergence timing was not evaluated in the study.

Author Response

General slug related questions

How wide should a slug plot be? Minimum of 4 rows of 10 feet to avoid inter-plot inference.

How long would you say between soil disturbance and an increase in slug population? Difficult to predict because re-colonization depends on weather conditions following soil disturbance. With favorable fall weather when eggs are laid, slug populations could rebound to economic levels during the spring following a light disking the previous fall.

 Specific questions to the manuscript:

May want to briefly mention iron phosphate baits in the introduction such as Sluggo, Iron Fist, and Ferroxx. I concede that efficacy is pretty variable compared to Deadline.  Added one sentence to mention the iron phosphate products.

Please include planting date for all experiments. Changed sentence in the general description of field sites to: Plots at all sites were no-till planted during early to mid-May…. Also, added the specific planting date for each field site described for each experiment.  

With the Urea applications, do you have temperature, relative humidity and date of application?  Added date of application but did not record specific RH and temperature readings.

Do you have an estimate for proportion of D. laeve? You mention timing of egg hatch of slugs related to seedling growth was likely a factor contributing to different levels of slug damage, with the 1992 field planted earlier and with ‘observations indicate peak egg hatch occurred after 2-leaf stage’. This is similar to an anecdote in Ron Hammond’s 1996 slug bait paper. Can you elaborate any more on this? Do you have data suggesting what date peak egg hatch occurred near? I start finding gray garden neonates around the first week of April, but D. laeve anytime in the spring and late winter.  Unfortunately, the species composition of the slug populations in each field site and year was not determined; only cursory observations indicated that gray garden slug was the predominate species.

Can you confirm that the highest slug density was 1994 WREC even with 2nd lowest slug injury to plants?  The highest density at 1994 WREC was based on visual counts at night averaged over a relatively high number of replicated untreated plots. It was the highest density recorded at night but night counts were not conducted at all site/years.

The methods indicate that plots were harvested with a Massey Ferguson, but there is no indication of yield data in the results. If you do not intend to record yield, perhaps it would be better to remove that line. Yields were measured a several sites but showed no significant treatment effects, which was mentioned in the results.

Any thoughts as to why in-furrow reduced slug damage? Better/faster plant growth? Were seed slots closed?  The slots were closed but not all ammonium nitrate was buried by the in-furrow application because a portion of the fertilizer was deposited on the surface around the seed slot. Thus, it is likely that the salt and acid effects of ammonium nitrate acted as a direct irritant and impeded slug activity.

In the discussion – rephrase the sentence “Our results show that better approach is to apply….after plants have already emerged.” A pre-plant vs post-emergence timing was not evaluated in the study.  Changed the wording from “Our results show …” to “Our results suggest …”

Reviewer 2 Report

Although this study was conducted many years ago, the results and their analyses are still relevant and even helpful to the contemporary measures in managing slugs. The field experiments during the three consecutive years provide practical information on the usages of row cleaner and starter fertilizer to prevent slugs from damaging no-till corns. It seems that the methodology is clearly described and the results are soundly analyzed. I think this manuscript looks suitable to be published as it is, only with a couple minor edits in style as listed below.

1. It would be good-looking if the color code of the bar graphs in figures is consistent. For example, the “No row cleaner” bars are shown in red in Figures 1 – 3, but the “Row cleaner” bars in gray, white and white, respectively. The fertilizer bars follow similar pattern. Although different colors might be used to indicate different years, multiple colors could even lead to a bit confusion.

2. Does the “Untreated control” in Figure 3 refer to “No starter fertilizer”?.

3. Isn’t it necessary to replace K2O with K2O (subscript 2)?

Author Response

  1. It would be good-looking if the color code of the bar graphs in figures is consistent. For example, the “No row cleaner” bars are shown in red in Figures 1 – 3, but the “Row cleaner” bars in gray, white and white, respectively. The fertilizer bars follow similar pattern. Although different colors might be used to indicate different years, multiple colors could even lead to a bit confusion. I agree and have changed the bar colors to be consistent with respect to the main treatment factors.
  1. Does the “Untreated control” in Figure 3 refer to “No starter fertilizer”?. Yes, I changed Untreated control in Figure 3 to No starter fertilizer.
  2. Isn’t it necessary to replace K2O with K2O (subscript 2)? Changed K2O with K2O at all mentions in the text and graphs.

Reviewer 3 Report

I believe the manuscript contains valuable information that should be shared. However I find the presentation of the figures (such as minor issues of lack of axis labels),  and the combination of the results and discussion difficult to sort what experiment from what year provided the specific results.  I think the manuscript should be considered but I can not make specific recommendations other than to have the result separated from the discussion at this point. Other important suggestions are also needed to help clarify what data was collected, how it was evaluated and how it supported the conclusions outlined in the summary.
I attached the comments I made directly to the manuscript.

Author Response

Reviewer 3

Row cleaners might not be well know to non-corn growers suggest with row cleaner devices to clear a swath of crop residue. Changed wording to describe more clearly the residue management devices.

 Calm windy? seems an oxymoron, suggest calm conditions. Changed to under calm winds.

 Suggested company name for fertilizer input and urea-based N. Reviewer comments several times regarding this issue. Unfortunately, the company is unknown at this point. We don’t agree that the source of each of these generic inputs is necessary for someone to repeat the experiment. Moreover, there seems to be no consistent requirement for company names provided in published papers in the journal.

 What was the general slug population? Not quite sure why the reviewer made this comment in the methods and materials section. We stated at the beginning of this paragraph that all fields had a history of slug activity and then later results we specifically provide slug density data on certain fields.

 Reviewer list several suggestions regarding the sampling methods described in the last paragraph of section 2.2 of the methods and materials. Here, we stated that the methods vary slightly among experiments and simply provided a general description of the sampling methods and damage grading system used for the 1992 and 1993 experiments. We deleted the sentence describing slug counts that were made during the night which was given later in section 2.3.

 Reviewer commented on the timing of bait applications about 2 weeks. And suggested providing the number of days after planting, and whether the application corresponded to weather, plant susceptibility, or noted slug activity? We added: Baits were applied only once after the majority of eggs had hatched but before extensive slug injury occurred on the seedlings. Each rate was delivered with ….

 Regarding the potash treatments, reviewer commented on whether counts were recorded on dead slugs, what area was searched? immediately adjacent to the row/touching the plant? some area outside? Density counts were recorded mainly on the number of live slugs, although some moribund slugs may have been counted. The area searched was stated based on 20 consecutive plants from the center row of each plot.

 Reviewer made several comments about the general results presented at the end of first paragraph of the results and discussion section. We added “based on visual estimates” to the last sentence to explain how the percent residue cover was determined. We don’t agree that this general discussion of the variable slug counts by day or mention of the highest slug density recorded at WREC (1994) needs to be expanded to the other site years at this place. We do prepare slug counts in later sections.

 Reviewer suggests including how our statement that the “row cleaner treatment removed more than 80% of the plant residue from a 20-25 cm wide zone on the surface of the seed row” was measured in the methods. This was a general observation based on visual estimates, which we added.